# Visual prognosis and complications of congenital ectopia lentis: study protocol for a hospital-based cohort in Zhongshan Ophthalmic Center

Pusheng Xu,[1] Kityee Ng,[1] Ling Jin,[1] Charlotte Aimee Young,[2] Siyuan Liu,[1] Xiaolin Liang,[1] Zhenzhen Liu [ID],[1] Xinyu Zhang,[1] Guangming Jin [ID],[1] Danying Zheng [ID][1]

PX and KN are joint first authors.

[1]State Key Laboratory of Ophthalmology, Zhongshan Ophthalmic Center, Sun Yat-Sen University, Guangzhou, Guangdong, China
[2]Albany Medical College, Albany, New York, USA

**Correspondence to**
Dr Danying Zheng;
zhengdyy@163.com

## ABSTRACT

**Introduction** Congenital ectopia lentis (CEL) is a rare ocular disease characterised by the dislocation or displacement of the lens. Patients with mild lens dislocations can be treated with conservative methods (eg, corrective eyeglasses or contact lenses). In contrast, patients with severe CEL usually require surgical management. However, few studies have focused on the visual prognosis and complications in conservative and surgical management of patients. This study aims to investigate the prognosis and complications in patients with CEL with conservative and surgical management, which is vital for CEL management, especially the choice of surgical timing and surgical method.

**Methods and analysis** A cohort study will be conducted at Zhongshan Ophthalmic Center. We plan to recruit 604 participants diagnosed with CEL and aged ≥3 years old. Patients with mild lens subluxation and stable visual conditions will be included in the non-surgical group and follow-up at 1, 2 and 3 years after enrolment. Patients with severe lens subluxation who accept CEL surgery will be included in the surgical group. Different surgical techniques, including phacoemulsification, in-the-bag intraocular lens implantation (with or without capsular tension ring) and trans-scleral fixation, will be used depending on the severity of dislocation. Patients will be followed up at 3 months, and 1, 2 and 3 years postoperatively. Over a 5-year follow-up period, patients will receive a detailed ocular examination, including optometry, biological measurement, specular microscopy, ultrasound biomicroscopy, anterior segment and posterior segment optical coherence tomography (OCT), OCT angiography, echocardiography and questionnaires on vision-related quality of life. The primary outcome is the change of best-corrected visual acuity and the incidence of complications in both groups.

**Ethics and dissemination** Ethics approval was obtained from the ethics committee of the Zhongshan Ophthalmic Center (number: 2022KYPJ207). Study findings will be published in a peer-reviewed journal.

**Trial registration number** NCT05654025.

## STRENGTHS AND LIMITATIONS OF THIS STUDY

⇒ This study is a cohort of congenital ectopia lentis (CEL) in the Chinese population with a relatively large sample size and a comparatively long (5 years) follow-up.

⇒ This study assesses important ophthalmic measurements with significant clinical implications, including visual acuity, intraocular pressure, axial length, high-order aberrations, choroidal thickness and choriocapillaris flow deficits.

⇒ Important systemic indicators of Marfan syndrome are measured, including the metacarpophalangeal joint length, incidence of valvular heart disease, aortic root diameter and Z-score.

⇒ In this study, CEL in age less than 3 years old is excluded and participants could be lost to follow-up after surgery.

⇒ This study focuses on individuals with CEL, but different subtypes of CEL may not be classified.

## INTRODUCTION

Congenital ectopia lentis (CEL) is a rare ocular disease characterised by the dislocation or displacement of the lens. Several systemic diseases have been reported to be associated with CEL, such as Marfan syndrome, homocystinuria, Weill-Marchesani syndrome and Ehlers-Danlos syndrome.[1] However, some patients with CEL may have no known systemic manifestations.[2]

Patients with mild lens dislocations can be treated with conservative methods (eg, corrective eyeglasses or contact lenses).[3] However, progressive subluxation or complete dislocation of the lens can cause a high degree of myopia or astigmatism, even amblyopia.[4] Surgical intervention is crucial in the management of patients with severe CEL.[5] Currently, there is no unified standard for operation timing for CEL. A previous study recommended surgical intervention when the near

vision of children was worse than 0.4 in LogMAR acuity.[6] However, some researchers suggested that surgery should be performed when the best-corrected visual acuity (BCVA) is less than 0.3 or monocular diplopia occurs.[4 7] Few studies have focused on the refractive change and visual prognosis in patients with CEL both with conservative management and surgical management.[8]

Several surgical techniques have been reported over the past decades, such as lensectomy, phacoemulsification without intraocular lens (IOL) implantation, phacoemulsification and IOL implantation (with or without capsular tension ring) and various trans-scleral fixations of IOL.[9–11] However, the safety and efficacy of these techniques have not been fully validated so far, especially in the Chinese population. Herein, we will conduct this cohort study at the Zhongshan Ophthalmic Center, one of the biggest ophthalmic hospitals in China.[12] All children diagnosed with CEL will be followed up for at least 3 years. Long-term changes in BCVA and the incidence of complications will be evaluated in patients with conservative and surgical management.

## METHODS AND ANALYSIS
This study will be conducted from 5 December 2022 to 5 December 2027. The study was registered on ClinicalTrials.gov (NCT05654025).

### Objective
The study is designed as a prospective clinical trial to investigate the prognosis and complications in patients with CEL with conservative and surgical management. The aim is to serve as a reference for disease management, specifically for choice of surgical timing and surgical method.

### Study design
An approximately 5-year cohort study will be conducted at Zhongshan Ophthalmic Center, Guangzhou, China. This study will adhere to the Declaration of Helsinki, and ethics have been approved by the ethics committee of the Zhongshan Ophthalmic Center (number: 2022KYPJ207). This study is designed used the Standard Protocol Items: Recommendations for Interventional Trials reporting guidelines.[13]

### Eligibility criteria
#### Inclusion criteria
1. Diagnosed with congenital lens dislocation and age ≥3 years old.
2. Agree to participate in this study with written informed consent from patients or legal guardians.

#### Exclusion criteria
1. History of ophthalmic trauma or other ophthalmic surgeries.
2. Combined with other ophthalmic diseases such as primary glaucoma, uveitis and corneal disease.
3. Patients who could not cooperate in the examinations.

### Study setting and participants
This cohort study will be conducted at Zhongshan Ophthalmic Center, one of the largest ophthalmic hospitals in China. This study aims to investigate the visual prognosis and complications in patients with CEL with different kinds of management. Patients aged 3 years or above diagnosed with CEL will be recruited from Zhongshan Ophthalmic Center. Moreover, each participant will be followed for at least 3 years.

Once the participants meet the requirement of our eligibility criteria, they will be asked to join the WeChat (an instant messaging tool) group on the phone and be provided with informed consent. Interested participants or their guardians will sign the consent form and, if appropriate, will complete a thorough ocular examination and systemic evaluations.

### Recruitment
The outpatient clinics will carry out the first screening at Zhongshan Ophthalmic Center. Potential participants will be further confirmed by eligibility and recruited at Zhongshan Ophthalmic Center for clinical trials. One of our researchers will contact the participants and explain the trial process in detail to ensure the participants or guardians fully understand the whole study. Once they agree and sign the consent form, further information will be provided, including the purpose of the study, examinations, the importance of follow-up time and duration, and possible risk in treatment. Then, the trial will proceed subsequently.

### Sample size
Sample size will be estimated as follows: assuming the incidence of complications in patients with CEL is 15%, and the margin of error is 20%. For a 5% significance level, $Z_{\alpha/2}$ is 1.96 for the two-tailed alternative hypothesis. Sample size$=(Z_{\alpha/2})^2 \times P(1-P) \times 1/E^2 = (1.96)^2 \times 15\% \times (1-15\%)/(15\% \times 20\%) = 544$. Assuming the loss ratio of 10%, the adjusted sample size will be $544/(1-0.1) = 544/0.9 = 604$.

### Preoperative management
All patients diagnosed with CEL will be followed up since the enrolment. The surgery will be performed when either of the following situations occurs: (1) BCVA <0.3 in LogMAR acuity; (2) monocular diplopia; (3) progressive subluxation of the lens affecting the pupillary axis; (4) complicated with severe cataract or secondary glaucoma or corneal endothelial decompensation or retinal detachment.[8]

### Preoperative medication
All patients will be routinely administered levofloxacin or tobramycin eye drops (four times a day for 3 days) before surgery to minimise the risk of infection. An intramuscular injection of ethamsylate (1 mg/kg)[14] will be used 30 min before surgery to reduce bleeding.

### Anaesthesia
General anaesthesia or retrobulbar anaesthesia will be used according to the standard clinical routine.

## Surgery methods management

The choice of surgical methods depends on the degree of ectopia lentis and the state of zonules. If the extent of the unhealthy zonules (broken or weak) is ≤180°, phacoemulsification and in-the-bag IOL implantation (with or without capsular tension ring) will be used. Otherwise, the capsular bag will be removed, and IOL will be fixed through trans-scleral fixation. Surgery will be performed by the same surgeon (DZ). Rayner 920H/970C or Sensar AR40e will be used as the implanted IOL.

The surgical techniques for patients who received in-the-bag IOL implantation are as follows: a 3.0 mm corneal tunnel incision will be made at 12 o'clock. Then, a continuous circular capsulorhexis will be performed manually. Iris hooks will stabilise the bag, and the lens will be aspirated with a phacoemulsifer. IOL will be implanted in the bag, and the capsular tension ring will be implanted when the IOL cannot be stably fixed.

For patients who received trans-scleral fixation of IOL, the surgical techniques will be performed as the previous study described.[8] In brief, a 3.0 mm corneal tunnel incision will be made at 12 o'clock. Then, a continuous circular capsulorhexis will be performed manually, and the capsular bag will be removed after the phacoaspiration. Trans-scleral fixation of IOL will be performed with the two IOL haptics sutured by 8-0 polypropylene at 2.0 mm posterior to the corneal limbus. Anterior vitrectomy will be performed when severe vitreous prolapse occurs.

## Postoperative management

After surgery, tobramycin and dexamethasone eye drops (four times a day) and ointment (once every night) will be routinely administered for 1 week. If intraocular pressure (IOP) is higher than 25 mm Hg, topical IOP-lowering medication will be used. If IOP is higher than 40 mm Hg, an intravenous drip of 20% mannitol will be used.[15] In case necessary, anterior chamber drainage will be performed through the side incision.

## Outcome measures

### Primary outcome

The primary outcome is the change of BCVA and the incidence of complications (time frame: non-surgical group is evaluated at the first visit and 1, 2, 3 years after enrolment; surgical group is evaluated at preoperation, 3 months, and 1, 2, 3 years postoperatively).

### Secondary outcomes

1. Change of axial length (time frame: non-surgical group is evaluated at the first visit and 1, 2, 3 years after enrolment; surgical group is evaluated at preoperation, 3 months, and 1, 2, 3 years postoperatively).
2. High-order aberrations (time frame: non-surgical group is evaluated at the first visit and 1, 2, 3 years after enrolment; surgical group is evaluated at preoperation, 3 months, and 1, 2, 3 years postoperatively).
3. Central cornea endothelial cell loss (time frame: preoperation, 3 months, and 1, 2, 3 years postoperatively).
4. The state of zonules (time frame: non-surgical group is assessed at the first visit and 1, 2, 3 years after enrolment; surgical group will be assessed preoperatively).
5. Anterior chamber angle (time frame: non-surgical group is assessed at the first visit and 1, 2, 3 years after enrolment; surgical group will be assessed preoperatively, 3 months, and 1, 2, 3 years postoperatively).
6. Tilt and eccentricity of IOL (time frame: 3 months, and 1, 2, 3 years postoperatively).
7. IOP (time frame: non-surgical group is evaluated at the first visit and 1, 2, 3 years after enrolment; surgical group is evaluated at preoperation, 3 months, and 1, 2, 3 years postoperatively).
8. Aortic root diameter (time frame: non-surgical group is evaluated at the first visit and 1, 2, 3 years after enrolment; surgical group is evaluated at preoperation and 1, 2, 3 years postoperatively).
9. Aortic root (sinuses of Valsalva) Z-score, adjusted by body surface area (Z-score) (time frame: non-surgical group is evaluated at the first visit and 1, 2, 3 years after enrolment; surgical group is evaluated at preoperation and 1, 2, 3 years postoperatively).
10. Incidence of valvular heart disease (time frame: non-surgical group is evaluated at the first visit and 1, 2, 3 years after enrolment; surgical group is evaluated at preoperation and 1, 2, 3 years postoperatively).
11. Body mass index (BMI) (time frame: non-surgical group is evaluated at the first visit and 1, 2, 3 years after enrolment; surgical group is evaluated at preoperation and 1, 2, 3 years postoperatively).
12. Metacarpophalangeal joint length (time frame: non-surgical group is evaluated at the first visit and 1, 2, 3 years after enrolment; surgical group is evaluated at preoperation and 1, 2, 3 years postoperatively).
13. Choroidal thickness (time frame: non-surgical group is evaluated at the first visit and 3 years after enrolment; surgical group is evaluated at preoperation and 3 years postoperatively).
14. Choriocapillaris flow deficits (time frame: non-surgical group is evaluated at the first visit and 3 years after enrolment; surgical group is evaluated at preoperation and 3 years postoperatively).
15. Genetic mutation state of patients (time frame: preoperation).
16. Vision-related quality of life (time frame: non-surgical group is evaluated at the first visit and 3 years after enrolment; surgical group is evaluated at preoperation and 3 years postoperatively).

## Examinations

Baseline data and follow-up examination items are as follows:

1. Demographic characteristics, including name, gender and date of birth.

2. Slit-lamp examination: slit-lamp examination (BQ-900, Haag Streit, Switzerland) and fundus examination will be performed at each visit.

3. Visual acuity and refraction: the uncorrected visual acuity and BCVA will be evaluated with Early Treatment of Diabetic Retinopathy Study (ETDRS) LogMAR visual acuity chart (Precision Vision, Villa Park, Illinois, USA) at a test distance of 4.0 m. The refractive error will be determined by subjective refraction following an objective measurement. Spherical equivalent will be obtained with the calculation of spherical power plus half of the cylindrical power.

4. Axial length measures: axial length will be measured using IOLMaster 700 (Carl Zeiss, Jena, Germany).

5. High-order aberrations will be assessed using Nidek OPD-Scan III (Gamagori, Japan).

6. Corneal endothelial cell counting and morphology will be detected using an endokeratoscope (SP-2000P, Topcon, Japan).

7. The state of zonules will be assessed clock by clock with ultrasound biomicroscopy. The clock range of zonular disruption or loosening will be recorded. These results will be confirmed using slit-lamp photography after dilation.

8. The structure of the anterior chamber angle will be examined using a Tomey Casia 2 anterior segment optical coherence tomography (OCT) (Tomey, Tokyo, Japan).[16]

9. The tilt and eccentricity of IOL will be evaluated using Pentacam AXL (Oculus, Germany).[17]

10. IOP measures: the IOP in the patient's eyes will be measured using a non-contact tonometer at each visit, and the average of the three measures will be taken.

11. Echocardiography examination: this examination will be performed using Doppler echocardiography (HP/Philips Sonos 5500, Philips, Bothell, Washington, USA). A skilled technician will measure the aortic root diameter. Z-score will be calculated using the Marfan foundation's calculator (https://marfan.org/dx/zscore-children/). Normal Z-score ranges from −2 to 2. A dilated aortic root is defined as a Z-score ≥2.0. A larger Z-score is associated with an increased risk of aortic complications such as dissection, rupture and valvular regurgitation. An experienced cardiologist will determine the presence or absence of heart valve disease.

12. BMI: BMI is a person's weight in kilograms divided by the square of height in metres. The normal range for the Chinese population is 18.5–23.9. A higher BMI can indicate higher levels of body fat.

13. Metacarpophalangeal joint length will be measured in hand radiograph.

14. The choroidal thickness and choriocapillaris flow deficits will be measured using posterior segment OCT and OCT angiography (Zeiss Cirrus 5000 with AngioPlex).

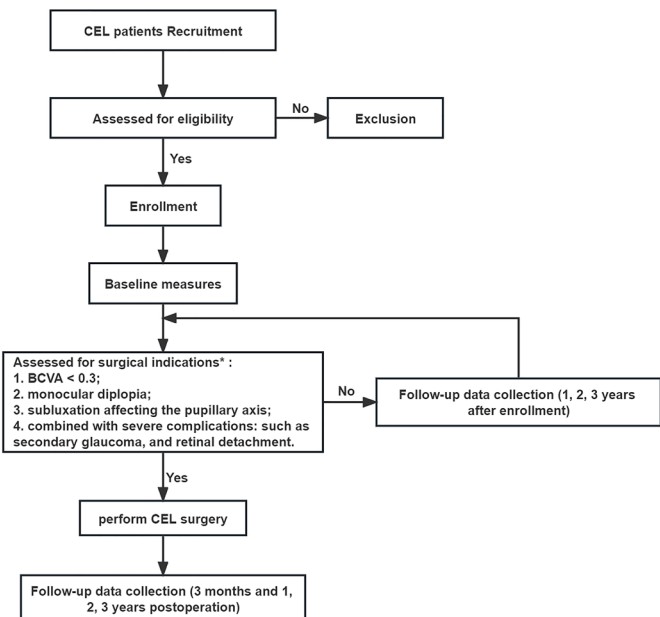

**Figure 1** Flow chart of the study. *The surgery will be performed when either of the following situations occurs. BCVA, best-corrected visual acuity; CEL, congenital ectopia lentis.

15. Gene detection: genomic DNA from each subject in the study will be analysed by whole-exome sequencing to detect mutations and diagnosis.

16. Vision-related quality of life will be assessed using the Pediatric Eye Questionnaire.[18]

17. Adverse events or complications will be collected at each visit.

## Workflow

The workflow will be carried out in the order mentioned above, starting with patients' recruitment and ending with a minimum of 3 years' follow-up (figure 1). The non-surgical group will be followed up 1, 2 and 3 years after enrolment. The surgical group will be followed up at 3 months, and 1, 2, 3 years postoperatively. The examinations planned for each follow-up time point are shown in online supplemental table 1.

## Data collection and management

The case report form (CRF) will collect basic demographic and clinical information. Results of eye examinations will be recorded on paper, and questionnaires will be filled out in the paper form. Original data will enter the lens dislocation case database at the Zhongshan Ophthalmic Center. After the study, relevant documents will be stored securely at the Zhongshan Ophthalmic Center for 10 years for specific scientific research purposes.

## Statistical analysis plan

Statistical analysis will be performed using Stata V.15.0 (StataCorp, College Station, Texas, USA). Quantitative data conforming to a normal distribution will be described as the mean±SD. The difference between the baseline and follow-up data will be compared by one-way analysis

of variance. The median±IQR will describe the quantitative data of the skewness distribution. The Wilcoxon signed-rank test will be used to compare these data's differences. For qualitative data described as proportions, the $X^2$ test or Fisher's exact test will be used to compare the differences between groups. P<0.05 is considered statistically significant. The differences and 95% CI in the changes of BCVA and other parameters will be calculated. The univariable linear regression model will estimate these changes and their associated factors. All variables with p<0.05 in the univariable regression analysis will be included in the multivariable linear regression model.

## Study monitoring

Clinical examiners will regularly check each patient's informed consent and eligibility to ensure that all CRFs are correct and in accordance with the original data. All errors and omissions should be recorded and corrected. The examiner should ensure every participant's withdrawal and loss of follow-up are recorded and explained in CRF, and all adverse events are recorded. The surgical operations in this project are standard methods and do not pose significant risks. The principal risks include general anaesthesia adverse effects and adverse drug reactions. This cohort study will be conducted under the guidance of the ethics committee of the Zhongshan Ophthalmic Center.

## Patient and public involvement

Neither the patients nor the public are involved in our research's design, conduct, reporting or dissemination plans.

## Ethics and dissemination

Ethics approval was obtained from the ethics committee of the Zhongshan Ophthalmic Center (number: 2022KYPJ207). Signed consent will be obtained from the legal guardians of participants after they have been informed of the study workflow and their right to withdraw from the cohort study. This project has been designed following the principles of the Declaration of Helsinki.

The content of this cohort study is confidential information. Clinical records and data sets will be kept at the Zhongshan Ophthalmic Center in strict confidence and will only be assessed by the study investigators and authorised personnel. The results without personal data will be disseminated through peer-reviewed publications and conference presentations.

**Contributors** PX, KN, LJ, ZL, XZ, GJ and DZ conceived and designed the study. PX, KN, SL and XL wrote the draft. CAY, GJ and DZ revised the draft. GJ will lead the statistical analysis. ZL, XZ, GJ and DZ will oversee data acquisition and implementation on-site. All authors reviewed and approved the final manuscript.

**Funding** This study was supported by the National Natural Science Foundation of China (81873673, 81900841) and the Basic and Applied Basic Research Foundation of Guangdong Province (2021A1515011673, 2022A1515011181).

**Competing interests** None declared.

**Patient and public involvement** Patients and/or the public were not involved in the design, or conduct, or reporting, or dissemination plans of this research.

**Patient consent for publication** Not required.

**Provenance and peer review** Not commissioned; externally peer reviewed.

**ORCID iDs**
Zhenzhen Liu http://orcid.org/0000-0002-4853-2474
Guangming Jin http://orcid.org/0000-0001-9994-6338
Danying Zheng http://orcid.org/0000-0003-1315-7130

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
