## [Reviewer comments · BMJ Open]

ARTICLE DETAILS

TITLE (PROVISIONAL)	Visual Prognosis and Complications of Congenital Ectopia Lentis: Study Protocol for a Hospital-Based Cohort in Zhongshan Ophthalmic Center
AUTHORS	Xu, Pusheng; Ng, Kityee; Jin, Ling; Young, Charlotte; Liu, Siyuan; Liang, Xiaolin; Liu, Zhenzhen; Zhang, Xinyu; Jin, Guangming; Zheng, Danying

VERSION 1 – REVIEW

REVIEWER	Betul Gunay University of Health Sciences Turkey, Turkey
REVIEW RETURNED	25-Feb-2023

GENERAL COMMENTS	This is a very well-designed and interesting study. I believe that it will make significant contributions to clinical practice.
---

REVIEWER	Eva Stifter University of Vienna, Dept of Ophthalmology
REVIEW RETURNED	07-Mar-2023

GENERAL COMMENTS	No further comments. Study protocol for a prospective clinical trial (5-year longitudinal cohort study) until 2027.
---

REVIEWER	Patrícia loschpe Gus Hospital de Clínicas de Porto Alegre
REVIEW RETURNED	02-Apr-2023

GENERAL COMMENTS	The protocol is perfectly written and described and the results will be of interest for medical education.
--

REVIEWER	Julius Oatts University of California San Francisco
REVIEW RETURNED	02-Apr-2023

GENERAL COMMENTS	Overall, this is a well-written study protocol. I would like the authors to provide more rationale for why they chose the specific age range for inclusion criteria. Also, the authors should specify if there is an upper limit for inclusion criteria. I have suggested a major revision to figure 2 to better communicate what study activities will occur at each study time point. I also have several small suggestions for changes to strengthen the manuscript outlined line by line below: Page 5 Line 22 – “treated with surgical treatment” is redundant – I would change this here (and throughout) to “surgical management”
--

	Line 33 – what does “follow-up cohort” refer to? Later, this is called a follow-up prospective study. The authors should clarify their terminology and maintain consistency in the manuscript (it is called “follow-up cohort,” “prospective follow-up cohort,” “prospective clinical trial,” “longitudinal cohort study”) – these should all be standardized Line 43 – is the surgery dictated by surgeon preference or study protocol? It may be useful to at least name the different surgical techniques in the abstract Line 50 – “endokeratoscope” should be changed to “specular microscopy” (an endokeratoscope is the instrument used to perform the diagnostic, not the diagnostic itself) Page 8 Line 13 – authors may consider mentioning CEL with no known systemic manifestations Line 22 – add reference to support this statement Line 26 – authors should mention that this is LogMAR acuity Line 47 – again, what is a prospective, follow-up cohort? Isn’t this redundant? Page 11 Line 13 – specify that this is LogMAR acuity Line 14 – how is “progressive subluxation” defined and how will this be objectively monitored in a standardized and auditable fashion? Line 24 – the authors should specify plans for patients with fluoroquinolone allergy Page 12 Line 5 – do the authors plan to use iris hooks or capsular retraction hooks? Line 30 – give that the authors specify the exact medications in the preoperative routine, the same should be done for postoperative Page 13 Line 11 – how will the state of the zonules be objectively assessed? Page 14 Examinations – it would be very helpful if the examinations are described in the same order as they are listed in the secondary outcomes section Page 15 Line 21 – more details are needed on how UBM will be used to assess the zonules Figure 1 – instead of “assessed for surgical indications,” the authors should list the criteria for surgery and then the branching will follow that if yes, the patient has surgery and if no, they are followed with observation Figure 2 – this figure needs to be changed to a table or other diagram that shows exactly which diagnostics will be performed at each study visit.
--	--

REVIEWER	Bangtao Yao Southeast University
REVIEW RETURNED	07-Apr-2023

GENERAL COMMENTS	It is a great honor for having me to review the manuscript
--

	(Manuscript ID bmjopen-2023-072542) entitled "Visual Prognosis and Complications of Congenital Ectopia Lentis: A Prospective Follow-up Cohort Study." The authors aim to investigate the prognosis and complications in CEL patients with conservative and surgical treatment. In my opinion, this study deserves to be published, with minor revision.  1.The limitations of the study were not described in the "Strengths and limitations of this study" section. (P7) 2.Regarding patients presented bilaterally, how to manage this condition in detail? (e.g., surgical timing, sequential or simultaneous?) 3.Ectopia lentis secondary to systemic diseases was progressive. (e.g., In homocysteinemia, it has been seen in 30% of patients until age 10, and more than 90% in their third decade.) If the extent of the unhealthy zonules is less than 180°in the early stage, and the capsular tension ring is not placed during the surgery, how to manage the progressive zonules? 4. Glaucoma is excluded in the exclusion criteria (P9, Line55), how about (P11, Line17)? 5.Regarding transscleral fixation of IOL, knotless transscleral suture fixation or traditional transscleral suture fixation? (PMID: 36243894). 6.The choice of different surgical timing seems to be unmentioned. 7.What if the anterior vitrectomy is unsuccessful for treating the lens moved into vitreous cavity? 8.Why is ethamsylate conducted? (P11, Line27). 9.Some of the cited references are outmoded.
--	---

VERSION 1 – AUTHOR RESPONSE

Reviewer: 1

Dr. Betul Gunay, University of Health Sciences Turkey, Turkey

Comments to the Author:

This is a very well-designed and interesting study. I believe that it will make significant contributions to clinical practice.

Response : Thank you for your favorable comments.

Reviewer: 2

Dr. Eva Stifter, University of Vienna

Comments to the Author:

No further comments. Study protocol for a prospective clinical trial (5-year longitudinal cohort study) until 2027.

Response : Thank you for your comments.

Reviewer: 3

Dr. Patrícia Ioschpe Gus, Hospital de Clínicas de Porto Alegre

Comments to the Author:

The protocol is perfectly written and described and the results will be of interest for medical education.

Response : Thank you for your favorable comment.

Reviewer: 4

Dr. Julius Oatts, University of California San Francisco

Comments to the Author:

Overall, this is a well-written study protocol. I would like the authors to provide more rationale for why they chose the specific age range for inclusion criteria. Also, the authors should specify if there is an upper limit for inclusion criteria. I have suggested a major revision to figure 2 to better communicate

what study activities will occur at each study time point. I also have several small suggestions for changes to strengthen the manuscript outlined line by line below:

Response : Thank you for your professional comments. Our previous research found that children aged ≤ 3 years old only account for 8.82% of the whole CEL population (PMID: 30225232), and these children are often difficult to finish the necessary examinations. Therefore, this study only included children older than 3 years old. In this study, congenital ectopia lentis patients with different ages were included and we did not set an upper age limit for inclusion criteria considering. We have modified Figure 2 into a table format to better show the examination items to be carried out during each study period now.

Page 5

Line 22 – “treated with surgical treatment” is redundant – I would change this here (and throughout) to “surgical management”

Response : Thank you for your comments. We have revised it to “surgical management” throughout the manuscript.

Line 33 – what does “follow-up cohort” refer to? Later, this is called a follow-up prospective study. The authors should clarify their terminology and maintain consistency in the manuscript (it is called “follow-up cohort,” “prospective follow-up cohort,” “prospective clinical trial,” “longitudinal cohort study”) – these should all be standardized

Response : Thank you for your comments. We have revised it to “cohort” throughout the manuscript.

Line 43 – is the surgery dictated by surgeon preference or study protocol? It may be useful to at least name the different surgical techniques in the abstract

Response : Thank you for your comments. The surgery is dictated by the study protocol. We have revised this part in the abstract now.

Line 50 – “endokeratoscope” should be changed to “specular microscopy” (an endokeratoscope is the instrument used to perform the diagnostic, not the diagnostic itself)

Response : Thank you for your comments. We have revised it accordingly.

Page 8

Line 13 – authors may consider mentioning CEL with no known systemic manifestations

Response : Thank you for your comments. We have revised it in the “introduction” section accordingly.

Line 22 – add reference to support this statement

Response : Thank you for your comments. We have added a reference to support this statement now.

Line 26 – authors should mention that this is LogMAR acuity

Response : We have revised it in the “introduction” section accordingly.

Line 47 – again, what is a prospective, follow-up cohort? Isn't this redundant?

Response : Thank you for your comments. We have revised it to “cohort” throughout the manuscript.

Page 11

Line 13 – specify that this is LogMAR acuity

Response : Thank you for your comments. We have revised it in the manuscript.

Line 14 – how is “progressive subluxation” defined and how will this be objectively monitored in a standardized and auditable fashion?

Response : Thank you for your professional comments. Currently, there is indeed no recognized, objective standard for the diagnosis of “progressive subluxation”(PMID: 32155956). Considering the severity of lens dislocation in congenital ectopia lentis may increase over time, “progressive subluxation” is defined as increasing degree of zonular disruption or loosening which could be detected by ultrasound biomicroscopy (UBM) and/or slit-lamp photography after dilation.

Line 24 – the authors should specify plans for patients with fluoroquinolone allergy

Response : Thank you for your comments. We have revised this in the manuscript. Tobramycin eye drops would be used in patients with fluoroquinolone allergy.

Page 12

Line 5 – do the authors plan to use iris hooks or capsular retraction hooks?

Response : Thank you for your comments. We plan to stabilize the bag by using iris hooks to hook onto the bag, since capsular retraction hooks are not available in our hospital. The following image is an intraoperative screenshot of us using an iris hook to stabilize the capsular bag.

Line 30 – give that the authors specify the exact medications in the preoperative routine, the same should be done for postoperative.

Response : Thank you for your comments. We have revised it accordingly now.

Page 13

Line 11 – how will the state of the zonules be objectively assessed?

Response : Thank you for your comments. The state of the zonules will be objectively assessed using ultrasound biomicroscopy (UBM) and slit-lamp photography after dilation.

Page 14

Examinations – it would be very helpful if the examinations are described in the same order as they are listed in the secondary outcomes section

Response : Thank you for your comments. We have reordered the examinations accordingly.

Page 15

Line 21 – more details are needed on how UBM will be used to assess the zonules

Response : Thank you for your comments. We have added more details information on how UBM will be used to assess the zonules now.

Figure 1 – instead of “assessed for surgical indications,” the authors should list the criteria for surgery and then the branching will follow that if yes, the patient has surgery and if no, they are followed with observation

Response : Thank you for your comments. We have revised the Figure 1 accordingly.

Figure 2 – this figure needs to be changed to a table or other diagram that shows exactly which diagnostics will be performed at each study visit.

Response : Thank you for your suggestion. We have revised the Figure 2 into Supplemental table 1 (According to the journal’s requirement: tables with more than 9 columns should be uploaded separately as a Supplemental Material file).

Reviewer: 5

Dr. Bangtao Yao, Southeast University

Comments to the Author:

It is a great honor for having me to review the manuscript (Manuscript ID bmjopen-2023-072542) entitled "Visual Prognosis and Complications of Congenital Ectopia Lentis: A Prospective Follow-up Cohort Study."

The authors aim to investigate the prognosis and complications in CEL patients with conservative and surgical treatment.

In my opinion, this study deserves to be published, with minor revision.

1. The limitations of the study were not described in the "Strengths and limitations of this study" section. (P7)

Response : Thank you for your comments. We have added limitations in the "Strengths and limitations of this study" section now.

2. Regarding patients presented bilaterally, how to manage this condition in detail? (e.g., surgical timing, sequential or simultaneous?)

Response : Thank you for your comments. As we mentioned in the manuscript in page 11. Patients who satisfied the indications of operation would accept surgical treatment. Otherwise, routine follow-up and conservative treatment will be continued. If both eyes meet the indications of operation, surgery would be performed on the worse eye first, followed by another eye.

3. Ectopia lentis secondary to systemic diseases was progressive. (e.g., In homocysteinemia, it has been seen in 30% of patients until age 10, and more than 90% in their third decade.)

If the extent of the unhealthy zonules is less than 180° in the early stage, and the capsular tension ring is not placed during the surgery, how to manage the progressive zonules?

Response : Thank you for your comments. For patients who do not have the capsular tension ring placed during surgery, follow-up would be more closely for them. If dislocation of the intraocular lens (IOL) was detected and indications for surgical treatment of IOL dislocation was met, IOL exchange or repositioning surgery would be performed.

4. Glaucoma is excluded in the exclusion criteria (P9, Line55), how about (P11, Line17)?

Response : Thank you for your comments. We have added more details about the exclusion criteria now (P9, Line55). Primary glaucoma is excluded in this study, but glaucoma secondary to ectopia lentis will not be excluded.

5. Regarding transscleral fixation of IOL, knotless transscleral suture fixation or traditional transscleral suture fixation? (PMID: 36243894).

Response : Thank you for your comments. Knotless transscleral suture fixation of IOL will be performed in the patients.

6. The choice of different surgical timing seems to be unmentioned.

Response : Thank you for your comments. Considering the timing of surgical intervention for ectopia lentis are still debated, in this study, surgery is indicated when either of the following situations occurs: 1) BCVA < 0.3 in LogMAR acuity; 2) monocular diplopia; 3) progressive subluxation of the lens affecting the pupillary axis; 4) complicated with severe cataract or secondary glaucoma or corneal endothelial decompensation or retinal detachment. (P11, Line 12-19)

7. What if the anterior vitrectomy is unsuccessful for treating the lens moved into vitreous cavity?

Response : Thank you for your comments. If the anterior vitrectomy is unsuccessful for treating the lens moved into vitreous cavity, a complete vitrectomy would be performed.(PMID: 28207607)

8. Why is ethamsylate conducted? (P11, Line27).

Response : Thank you for your comments. Most of our patients with CEL are children. Children have abundant conjunctival blood vessels and we need to open the bulbar conjunctiva during surgery. Ethamsylate is a synthetic hemostatic drug indicated in cases of capillary bleeding. Ethamsylate will be used in this cohort for it has been suggested as a hemostatic agent in surgical or postsurgical capillary bleeding in several well-controlled clinical trials. (PMID: 33064669; PMID: 16772766)

9. Some of the cited references are outmoded.

Response : Thank you for your comments. We have updated the references now.

VERSION 2 – REVIEW

REVIEWER	Julius Oatts University of California San Francisco
REVIEW RETURNED	07-Jun-2023
GENERAL COMMENTS	The authors have appropriately addressed all concerns.